# Regulation of Skin Barrier Function via Competition between AHR Axis versus IL-13/IL-4‒JAK‒STAT6/STAT3 Axis: Pathogenic and Therapeutic Implications in Atopic Dermatitis

**DOI:** 10.3390/jcm9113741

**Published:** 2020-11-20

**Authors:** Masutaka Furue

**Affiliations:** 1Department of Dermatology, Graduate School of Medical Sciences, Kyushu University, Fukuoka 812-8582, Japan; furue@dermatol.med.kyushu-u.ac.jp; Tel.: +81-92-642-5581; Fax: +81-92-642-5600; 2Research and Clinical Center for Yusho and Dioxin, Kyushu University Hospital, Fukuoka 812-8582, Japan

**Keywords:** atopic dermatitis, interleukin-13, filaggrin, JAK, STAT6, aryl hydrocarbon receptor, OVOL1, NRF2, ROS, skin barrier

## Abstract

Atopic dermatitis (AD) is characterized by skin inflammation, barrier dysfunction, and chronic pruritus. As the anti-interleukin-4 (IL-4) receptor α antibody dupilumab improves all three cardinal features of AD, the type 2 cytokines IL-4 and especially IL-13 have been indicated to have pathogenic significance in AD. Accumulating evidence has shown that the skin barrier function is regulated via competition between the aryl hydrocarbon receptor (AHR) axis (up-regulation of barrier) and the IL-13/IL-4‒JAK‒STAT6/STAT3 axis (down-regulation of barrier). This latter axis also induces oxidative stress, which exacerbates inflammation. Conventional and recently developed agents for treating AD such as steroid, calcineurin inhibitors, cyclosporine, dupilumab, and JAK inhibitors inhibit the IL-13/IL-4‒JAK‒STAT6/STAT3 axis, while older remedies such as coal tar and glyteer are antioxidative AHR agonists. In this article, I summarize the pathogenic and therapeutic implications of the IL-13/IL-4‒JAK‒STAT6/STAT3 axis and the AHR axis in AD.

## 1. Introduction

Atopic dermatitis (AD) is a common inflammatory skin disease that accounts for almost 10% of dermatologic patients in Japan [1]. Its prevalence is 10% to 16.5% in children under 5 years old, which showed a tendency to increase from the 1980s to the early 2000s globally [2,3]. The patients with AD show heterogeneous clinical presentation and at least six subtypes are defined by their natural course with the early-onset-early-resolving type being the most frequent one [4,5]. However, subsequent recurrence is also frequently seen and the recurrent symptoms are usually more indolent than the initial ones [4,5]. AD is characterized by skin inflammation, skin barrier dysfunction, and chronic pruritus, which reduce quality of life and treatment satisfaction among afflicted patients [6,7,8]. Their adherence to treatment is also markedly deteriorated [9,10,11]. Skin barrier dysfunction causes frequent colonization of *Staphylococcus aureus*, which can lead to further deterioration of the barrier function [12,13]. Although not as conspicuously as in psoriasis, AD is known to be associated with systemic inflammatory diseases such as cardiovascular diseases and metabolic syndrome [14].

Since the discovery of T helper 2 (Th2) cells by Mosmann et al. [15], it was proposed that interleukin (IL)-13 and IL-4 play critical roles in the pathogenesis of AD [16,17,18,19]. This is underscored by the fact that clinical symptoms and pruritus in AD are significantly improved by the anti-IL-4 receptor α (IL-4Rα) antibody dupilumab, which inhibits both IL-13 and IL-4 signaling [20,21,22]. In addition to clinical improvement, dupilumab has been proven to normalize the elevated levels of cytokines and chemokines, which are related to cutaneous inflammation and pruritus in AD [22]. Dupilumab also restores the decreased levels of barrier-related proteins such as filaggrin (FLG) and loricrin (LOR) [22]. Therefore, IL-13/IL-4 signaling is now considered to be the essential core of the pathogenesis of AD [23,24,25]. The purpose of this review article is to discuss the pathogenesis and treatment of AD by highlighting the regulatory mechanisms of skin barrier-related molecules.

## 2. IL-13/IL-4 Signaling and AD

The receptor system for IL-13/IL-4 differs between hematopoietic and non-hematopoietic peripheral cells [25] (Figure 1). In lymphocytes and dendritic cells, IL-4 signals through the IL-4Rα/γC complex. IL-4 binds IL-4Rα/γC and activates the downstream signaling molecules Janus kinase 1 (JAK1)/JAK3 and then signal transducer and activator of transcription (STAT)6. Activation of the IL-4‒IL-4Rα/γC‒JAK1/JAK3‒STAT6 axis induces Th2-deviated T-cell differentiation, IgE production in B cells, and Th2 chemokine production such as CCL17 and CCL22 from dendritic cells [25,26]. On the other hand, keratinocytes express IL-4Rα/IL-13Rα1 complex (Figure 1). Both IL-13 and IL-4 bind IL-4Rα/IL-13Rα1 and activate downstream JAK1/TYK2/JAK2 and then STAT6/STAT3. Activation of the IL-13/IL-4‒IL-4Rα/IL-13Rα1‒JAK1/TYK2/JAK2‒STAT6/STAT3 axis down-regulates FLG expression, disrupts the skin barrier function, and up-regulates the production of thymic stromal lymphopoietin (TSLP), IL-25, and IL-33 in keratinocytes [25,27,28]. IL-13 and IL-4 exert similar biological responses via IL-4Rα/IL-13Rα1 [29,30,31]. However, IL-13- and IL-4-producing cells are not the same. Lymph node T follicular helper (Tfh) cells produce IL-4 but not IL-13 [32]. A recent study revealed that IL-4 produced from Tfh cells essentially regulates IgE production [33]. In contrast, group 2 innate lymphoid cells (ILC2s) produce IL-13, but little IL-4 [34,35].

IL-13/IL-4 signatures are common in the lesional skin of AD relative to the level in healthy control skin [17,18,19]. Recent transcriptomic analyses have revealed that in AD, gene expression levels of IL-13 correlate more with the intensity of skin inflammation than those of IL-4 [22,36], suggesting that the IL-13-producing dermal ILC2s may be more involved in the pathogenesis of AD [35].

In AD, type 2-prone immune deviation is more prominent in lesional than in non-lesional skin, as well as in chronic lesions rather than acute ones [17,18]. Type 2 immune deviation has been reported to occur among circulating Th cells [37]. Skin homing T cells express cutaneous lymphocyte antigen (CLA). IL-13-producing CLA+ Th cells have been shown to be increased in the peripheral blood in pediatric and adult patients with AD [37]. Another study showed that IL-4-responsive T-cell proliferative reaction is also elevated in AD [38].

The expression of type 2 chemokines such as CCL17, CCL18, CCL22, and CCL26 is up-regulated in the lesional skin of AD [16,22]. CCL17, CCL18, and CCL22 are known to be produced from dendritic cells and dermal fibroblasts stimulated with IL-13/IL-4 and are chemoattractive for Th2 cells [16,22,26,39]. CCL26 is an eosinophil chemoattractant produced from IL-13/IL-4-treated endothelial cells [16,22,40]. The increased levels of these type 2 chemokines are down-regulated by the use of dupilumab or topical steroid to interfere with IL-4Rα or to reduce the production of IL-13/IL-4 [22,41].

The serum levels of CCL17 and CCL22 are elevated in AD and are well correlated with the severity of this disease [42,43]. Circulating squamous cell carcinoma antigen 2 (SCCA2, SERPINB4), one of the target gene products of IL-13/IL-4, can be used as a serum biomarker for AD [44,45,46]. Cytokine/chemokine profiling of interstitial fluids revealed significant elevations of IL-13 and CCL17 in the lesional dermis of AD compared with the levels in healthy controls [47]. Moreover, large amounts of CCL17 and CCL22 are known to be present in the tape-stripped cornified layer [48,49]. IL-5 is also a Th2 cytokine and potently induces the proliferation, differentiation, and chemotaxis of eosinophils [50]. Expression of the IL-5 gene has been shown to be increased in the lesional skin of pediatric and adult AD [17,18,19].

In addition to IL-4Rα/IL-13Rα1, keratinocytes also express IL-13Rα2 (Figure 1). IL-13Rα2 is a high-affinity receptor for IL-13, but it does not signal. Therefore, IL-13Rα2 serves as a decoy receptor and decreases the concentration of IL-13 in the microenvironment [51,52]. Notably, IL-13 does not alter the expression levels of IL-4R and IL-13Rα1, but it up-regulates the expression of IL-13Rα2 in keratinocytes [51]. Scratch injury has also been found to enhance the expression of IL-13Rα, but not IL-4Rα and IL-13Rα1, in keratinocytes [51]. Thus, IL-13 promotes the development of atopic inflammation via IL-4Rα/IL-13Rα1, but simultaneously triggers a negative feedback signal via the decoy receptor IL-13Rα2, which lowers excess amounts of extracellular IL-13 [51]. Pruritus-mediated scratching appears to exacerbate dermatitis [53]. However, scratch injury on keratinocytes also results in the up-regulation of IL-13Rα2 expression and forms another negative feedback circuit to inhibit excess IL-13 activity [51,52].

## 3. Role of IL-31 and IL-13/IL-4 in Atopic Pruritus

Pruritus is the major subjective symptom in AD [54]. Histamine is the essential pruritogen of urticaria and immediate-type allergic response, for which the first-line treatment is H1 receptor antagonists [55,56,57]. However, the anti-pruritic efficacy of H1 antagonists is quite limited in AD [58].

IL-31 is a type 2 cytokine produced from Th2 cells [59,60]. Its expression is elevated in the lesional skin of AD [22]. IL-31 induces elongation and branching of sensory nerve fibers [61,62]. Notably, the administration of IL-31 evokes scratching behavior in mouse, dog, and monkey, as well as pruritus in human [59]. The transcription factor hypoxia-inducible factor-2α (HIF-2α), also called endothelial Per–Arnt–Sim (PAS) domain protein 1 (EPAS1), is required for the production of IL-31 [60]. Binding of IL-31 to IL-31-sensitive dorsal root ganglion cells induces the production of neurokinin B and activates its neurokinin 3 receptor, which then triggers the production of an itch mediator, gastrin-releasing peptide [63]. A recent study also revealed a significant role of P2X3 receptor-positive nerves in chronic pruritus [64]. In addition, chronic pruritus is known to be associated with STAT3-dependent astrogliosis in the spinal dorsal horn [65].

Injection of the anti-IL-31 receptor A (IL-31RA) antibody nemolizumab significantly reduces the pruritus of patients with AD [66,67]. The monthly administration of nemolizumab for 52 weeks was also found to continuously inhibit the pruritus of AD [68]. Moreover, cotreatment with nemolizumab and topical steroid was reported to significantly augment the anti-pruritic effects of topical steroid monotherapy [69]. The anti-IL-31 antibody lokivetmab is now commercially available for the treatment of canine AD [70]. This highlights the pivotal role of IL-31 in atopic pruritus.

Murine and human sensory nerves express IL-4Rα/IL-13Rα1 [71]. Although IL-13/IL-4 have been reported not to induce acute pruritus [71], a recent study proved that they act as pruritogens and induce scratching behavior [72]. In addition, IL-13/IL-4 potentiate the pruritic response mediated by histamine or IL-31 [71]. IL-31 binds to IL-31RA and oncostatin M receptor (OSMR) heterodimer, and then activates JAK1/JAK2 and subsequently STAT3 (and STAT1 and STAT5) [59]. As IL-13/IL-4 activate JAK1–STAT6, JAK1 inhibitor reduces pruritus mediated by IL-31 or IL-13/IL-4 [71].

Colonization of *Staphylococcus aureus* up-regulates the expression of antimicrobial peptides such as human β-defensin 2 and -3 and RNAse7; however, IL-31 inhibits their expression and may accelerate the Staphylococcal infection [73]. Staphylococcal superantigen may also induce glucocorticoid insensitivity by inhibiting the nuclear translocation of glucocorticoid receptor α [74].

## 4. Regulation of Skin Barrier Function by Competition between AHR Axis and IL-13/IL-4‒JAK‒STAT6/STAT3 Axis

Under physiological conditions, homeostasis of the skin barrier function is regulated by the coordinated expression of barrier-related proteins, intercellular lipids, and corneodesmosomes in the granular and cornified layers [75,76]. The genes encoding many barrier-related proteins such as *FLG*, *LOR*, and involucrin (*IVL*) are encoded in the epidermal differentiation complex (EDC) region located on chromosome 1q21.3 [76]. The EDC region includes members of the cornified envelope precursor gene family such as *LOR* and *IVL*, the S100A protein gene family such as *S100A7* and *S100A8*, and the fused gene family such as *FLG* and *FLG2*, with this latter family being derived from fusion of the cornified envelope precursor gene and the S100A protein gene [76] (Figure 2).

Among various transcription factors, aryl hydrocarbon receptor (AHR) is essential to the coordinated up-regulation of EDC genes [25,77,78,79]. Several endogenous and exogenous ligands up-regulate the expression of FLG, LOR, and IVL [5,77,78,79] (Figure 3). For example, photoproducts produced by ultraviolet rays [80,81,82,83], bioproducts from commensal cutaneous microbiomes such as *Malassezia* and *Staphylococcus epidermidis* [84,85], and bioproducts from intestinal microbiomes [86] activate AHR and up-regulate the expression of FLG. Phytochemicals used in folk medicine [87,88,89,90,91] and some cosmetic ingredients [30] contain AHR ligands and up-regulate FLG expression. Medicinal coal tar and soybean tar glyteer are AHR ligands that increase FLG expression [31,92] (Figure 3).

Ligation of AHR induces its cytoplasmic-to-nuclear translocation and the expression of genes encoding *FLG*, *LOR*, *IVL*, and other barrier-related proteins in the EDC loci [25,77,78,79] (Figure 3). In addition to the AHR-mediated direct up-regulation of barrier-related proteins, the activation of AHR up-regulates the gene and protein expression of OVO-like 1 (OVOL1) transcription factor [81,90]. Cytoplasmic OVOL1 then translocates into the nucleus and further up-regulates the gene and protein expression of FLG and LOR [81,90]. However, OVOL1 is not involved in IVL up-regulation [90]. Activation of AHR simultaneously induces the expression of xenobiotic-metabolizing enzymes such as cytochrome P450 1A1 (CYP1A1), which can potentially metabolize AHR ligands [80]. Therefore, the activation of AHR by ligands is self-limiting, ensuring constitutive homeostatic regulation [80].

In the lesional skin of AD, the expression of FLG is down-regulated compared with that in healthy control skin [81,92]. Transepidermal water loss (TEWL) is significantly increased and the skin hydration is decreased in AD compared with the levels in healthy individuals [93]. Serum CCL17 levels are positively correlated with TEWL and negatively correlated with skin hydration in AD [93]. Topical steroids improve the skin inflammation of AD and normalize the increased levels of TEWL and decreased expression of FLG and LOR [41].

Genome-wide association studies (GWASs) have identified more than 30 genes conferring susceptibility to AD [94,95,96]. A meta-analysis of GWASs for those of European, Chinese, or Japanese ancestry revealed that the top three highly significant susceptibility genes are *FLG*, *OVOL1*, and *IL13* [96]. Among these, loss-of-function mutation of *FLG* is most significantly associated with the development of AD [75,96]. However, the majority of patients with AD do not exhibit loss-of-function mutation of *FLG* [97]. In addition, the loss-of-function mutation of *FLG* is less common in Southern Europe than Northern Europe [97], and is not found in some parts of Africa [98,99]. On subtropical Ishigaki Island, the prevalence of pediatric AD is lower than that in mainland Japan [100] and the loss-of-function mutation of *FLG* is not associated with the development of AD on this island [101].

Down-regulation of barrier-related proteins by IL-13/IL-4 is probably more significantly involved in the development of AD than loss-of-function mutation of *FLG* [31,81,89,90,92,102]. A recent report from Croatia revealed that loss-of-function mutation of *FLG* was detected in only 4 of 91 AD patients and none of 47 non-AD controls; it also showed that elevated TEWL is associated with skin inflammation but not with FLG mutation [103].

In epidermal keratinocytes, IL-13/IL-4 bind IL-4Rα/IL-13Rα1 heterodimer and activate downstream JAK1/TYK2/JAK2 and then STAT6/STAT3 [25] (Figure 1). Activation of the IL-13/IL-4‒JAK‒STAT6/STAT3 axis inhibits the AHR-mediated up-regulation of FLG, LOR, and IVL [31,81,90,92]; meanwhile, activation of the AHR axis inhibits the IL-13/IL-4-mediated STAT6 phosphorylation and restores the IL-13/IL-4-mediated FLG decrease [92] (Figure 4). In addition, activation of the IL-13/IL-4‒JAK‒STAT6/STAT3 axis inhibits the cytoplasmic-to-nuclear translocation of OVOL1 and inhibits the expression of FLG and LOR [81,90,102] (Figure 4). Moreover, the IL-13-induced STAT6 activation induces keratinocytes to produce periostin (Figure 1), and then periostin up-regulates the IL-24 expression and IL-24 inhibits the expression of FLG via STAT3 activation [104,105] (Figure 5). These results indicate that the IL-13/IL-4‒JAK‒STAT6/STAT3 axis affects several signaling pathways to inhibit the expression of barrier-related proteins. In parallel with this, upon IL-4 treatment, the permeability barrier of the cultured keratinocytic sheet is disrupted [106] and the distribution of cell surface E-cadherin is altered [107].

Given that the expression levels of barrier-related proteins are regulated via competition between the AHR axis and the IL-13/IL-4‒JAK‒STAT6/STAT3 axis [25,31,81,92], STAT6-deficient conditions may enhance the AHR axis and potentiate the skin barrier function. Findings have suggested that this is indeed the case. The skin barrier function in STAT6-deficient mice is significantly up-regulated compared with that in wild-type mice, as demonstrated by decreased TEWL, increased water content, decreased pH, decreased permeability of Evans blue, and increased LOR and FLG expression [108].

It is also known that IL-20, IL-22, IL-25, IL-31, and IL-33 can potentially reduce the expression of FLG, although the inhibitory mechanisms behind this are not well understood [109,110,111,112,113]. In addition, it has been reported that IL-20 and IL-24 are involved in the IL-31-mediated inhibition of FLG expression [110].

## 5. Skin Barrier Dysfunction Stimulates Keratinocytes to Produce TSLP, IL-25, and IL-33 and Promotes Type 2 Immune Deviation

Excessive activation of the IL-13/IL-4‒JAK‒STAT6/STAT3 axis and subsequent down-regulation of barrier-related proteins are likely to cause skin barrier dysfunction and the development of AD. On the other hand, it is known that epidermal keratinocytes in barrier-disrupted skin promote type 2 immune deviation. Epicutaneous application of hapten and mite antigens to barrier-disrupted skin was found to induce increased production of IL-4 and IgE in the regional lymph nodes compared with the levels in control mice with these antigens applied on barrier-intact skin [114]. Keratinocytes obtained from tape-stripped epidermis were also shown to produce larger amounts of CCL17, CCL22, and CCL5 and chemoattract IL-4-producing immune cells and eosinophils compared with the findings in their counterparts with intact epidermis [115]. Moreover, mice kept in dry air conditions show dry skin, with increases in the number of dermal mast cells and histamine concentration [116]. The application of moisturizer was also found to improve the dry skin and normalize the dermal histamine concentration [116]. In normal healthy individuals, decrease of skin water content significantly elevates serum CCL17 levels [93]. These results suggest that dry skin or skin barrier dysfunction can potentiate type 2 immune deviation in mammals.

The barrier dysfunction-induced type 2 immune deviation may be attributable to pro-type 2 cytokines such as TSLP, IL-25, and IL-33 produced from barrier-disrupted epidermis [17,117,118,119,120]. Skin with barrier disruption due to tape-stripping shows an increased level of TSLP [121]. TSLP stimulates murine dendritic cells to express OX40L and OX40L-positive dendritic cells induce OX40-positive T-cell differentiation [121,122,123,124]. OX40-positive T cells include a large number of Th2 cells expressing IL-4, IL-5, and IL-13 [121,122,123,124]. Therefore, OX40L/OX40 ligation is an essential checkpoint for promoting Th2-cell differentiation [123,124,125]. Human dendritic cells treated with TSLP also tend to induce a Th2-cell population producing IL-4, IL-5, and IL-13 [120]. TSLP was also found to up-regulate the production of CCL17 and CCL22 from human dendritic cells [120].

IL-25 (also called IL-17E) is a member of the IL-17 family [126]. IL-25-transgenic mice exhibit an increased number of blood eosinophils, elevated serum IgE levels, and the hyperproduction of IL-4, IL-5, and IL-13 [127]. The intranasal administration of IL-25 induces pulmonary hypereosinophilia and enhanced expression of IL-4, IL-5, IL-13, and eotaxin [128]. IL-25 also stimulates DCs to express OX40L and results in Th2 differentiation [129]. 

IL-33 is a member of the IL-1 family that is produced from peripheral tissues and induces type 2-dominant immune deviation [130]. It is overexpressed in keratinocytes derived from tape-stripped, barrier-disrupted epidermis [131]. The expression of IL-33 is also up-regulated in keratinocytes with herpes virus infection [132]. Therefore, the rapid exacerbation of AD in Kaposi’s varicelliform eruption may be attributable to IL-33 overexpression [132]. House dust mites also increase the production of IL-25 and IL-33 by keratinocytes via toll-like receptor 6 activation [133]. Many human allergens, such as those of house dust mites, fungi, and pollen, exhibit protease activity [130,134,135]. IL-33 is susceptible to the protease activity of these allergens, generating short-chain IL-33. Notably, this short-chain IL-33 shows much stronger biological activity than its original long-chain counterpart [130]. IL-33 also stimulates ILC2s to express OX40L and up-regulates the Th2 differentiation [125] (Figure 5). IL-25 can also enhance OX40L expression in ILC2s, but its potency is lower than that of IL-33 [125]. On the other hand, TSLP up-regulates the OX40L expression in dendritic cells, but not in ILC2s [122,123,124,125]. IL-33 also stimulates DCs to initiate type 2-prone immune deviation [136].

In a murine *Schistosoma mansoni* infection model, individual knockout treatment of each of *Tslp*, *Il25*, and *Il33* was shown not to interfere with IL-13/IL-4 production and subsequent fibrosis, whereas simultaneous knockdown of *Tslp*, *Il25*, and *Il33* was found to block the IL-13/IL-4 production and subsequent fibrosis [137]. These results suggest mutually overlapping or redundant bioactivity among TSLP, IL-25, and IL-33 [137]. Recent clinical trials have revealed that the anti-TSLP antibody tezepelumab is not efficacious for AD [138]. However, a single injection of the anti-IL-33 antibody ANB020 attenuates the skin symptoms in all 12 patients with AD [139]. These results may further stress the pivotal role of the IL-33–ILC2 axis in the pathogenesis of AD.

Figure 5 is a simplified scheme on the pathogenesis of AD. AD is actually quite a heterogeneous skin inflammation, so this scheme may not represent all of its aspects. However, it is now clear that the excessive production of IL-13, but probably not IL-4, in the skin activates the IL-13/IL-4‒JAK‒STAT6/STAT3 axis, down-regulates the expression of barrier-related proteins, induces barrier dysfunction, up-regulates the production of TSLP/IL-25/IL-33, and further accelerates type 2 immune deviation. The pruritogenic type 2 cytokines IL-31 and IL13/IL-4 stimulate sensory nerves and induce chronic pruritus. Pruritus evokes mechanical scratch, which further disrupts the barrier function and drives a vicious cycle toward type 2 immune deviation.

## 6. Th17/Th22 Cells and Chronicity in AD

Excessive IL13/IL-4 signaling is the critical driver in the pathogenesis of AD, while IL-17A-producing Th17 cells are present in the lesional skin of AD [36,140,141,142]. IL-22 produced from Th17 cells and Th22 cells is also detected at high levels in lesional skin compared with that in non-lesional skin and in chronic rather than acute lesions in AD [17,18,37]. The involvement of IL-17A has also been reported to be conspicuous in AD in Asians [141,142]. Moreover, serum IL-22 levels are significantly associated with the serum levels of CCL17 [143]. It has also been shown that the number of IL-22-producing CLA-positive Th cells is more increased in adult patients with AD than in pediatric ones [37]. The expression of IL-22 likely overwhelms IL-17A expression in the lesional skin of AD [36].

IL-22 increases the proliferative activity of keratinocytes and has been proposed to be involved in the chronicity of AD [144]. The anti-IL-22 antibody fezakinumab is more efficacious in severe than in mild AD [145] and in patients with high rather than low serum IL-22 levels [146]. Moreover, the decreases of IL-13 and IL-22 expression by topical steroid and tacrolimus are associated with the improvement of AD lesions [41,147,148]. These results suggest the role of IL-22 in AD chronicity [145]. In addition, some patients actually manifest psoriasiform eruption with IL-23/IL-17A overexpression during dupilumab therapy, suggesting that IL-23/IL-17A axis is potentially active and meaningful in these particular AD patients [149]. Multipolarity of these cytokines in AD may explain the heterogeneity and age/race differences of AD [141]. On the other hand, blockade of the IL-13/IL-4 signal by dupilumab alone was shown to normalize the increased IL-17A/IL-22 signals including S100A proteins, elafin and IL-23p19 [22], which may support the notion that excessive IL-13/IL-4 signal also causes the elevation of IL-17A/IL-22.

Th17/Th22 cells express CCR6 [150]. CCL20 is the only chemokine to recruit CCR6-positive immune cells [150]. Although epidermal keratinocytes are an abundant source of CCL20, its production is not influenced by IL-13/IL-4 signal [151]. However, the production of CCL20 is rapidly and significantly induced by epidermal scratch injury [152,153], suggesting that itch-scratch behavior itself may trigger CCL20 production to recruit Th17/Th22 cells [151,154,155]. The recruited Th17/Th22 cells may enhance their accumulation because IL-17A further stimulates keratinocytes to produce CCL20 [153]. It is possible that the blockade of itch-scratch behavior by dupilumab may attenuate the infiltration of Th17/Th22 cells.

IL-17A binds two heterodimeric receptors, IL-17 receptor A (IL-17RA)/IL-17RC and IL-17RA/IL-17RD, and activates downstream ACT1/TRAF6/CARMA2 signal complexes, NF-κB, and MAPKs [156,157]. IL-17A itself does not directly activate JAK-STAT pathways [158], but may activate STAT3 via IL-19 production [159]. IL-17A can enhance the expression of IVL and FLG2, and may not directly induce skin barrier dysfunction [25,160]. In parallel with this, the anti-IL-17A antibody secukinumab is not efficacious for treating AD [161]. On the other hand, IL-22 binds IL-22R1/IL-10R2 heterodimers and activates downstream JAK1/TYK2 and STAT3 [158,162]. It also inhibits the expression of IVL, LOR, and FLG [163,164,165]. Therefore, IL-22 is likely to be more responsible for the development of AD than IL-17A.

Th1 signatures such as interferon-γ (IFN-γ) are also detected more often in chronic than in acute lesional skin of AD [16,17,18]. IFN-γ binds a heterodimeric receptor, IFN-γ receptor I (IFNGR1) and IFN-γ receptor II (IFNGR2), and activates downstream JAK1/JAK2 and STAT1 [166]. Conflicting results have been reported on the effects of IFN-γ on the regulation of FLG expression: down-regulation [165] or up-regulation [167]. In our study, IFN-γ was shown to up-regulate the epidermal permeability barrier [106]. It is well known that IFN-γ inhibits Th2 cell differentiation and reduces the production of IL-14 and IL-13 [168,169]. The Th1 cell infiltration may be a compensatory reaction to attenuate excessive type 2 deviation in AD.

Another potential explanation for Th1 and Th17/Th22 cell infiltration in chronic AD is endothelin 1. Keratinocytes are an abundant source of endothelin 1 [170]. Endothelin 1 is abundantly expressed in basal keratinocytes under physiological conditions [119,171], but it is variably overexpressed in inflamed epidermis [119,172]. Endothelin 1 is one of the pruritogenic cytokines and has been confirmed to induce pruritus in mouse and human [173]. As mentioned above, dendritic cells treated with TSLP, IL-25, or IL-33 shift the immune response toward type 2 dominance [122,123,124,125,136]. In sharp contrast to this, DCs treated with endothelin 1 prompt T cells to differentiate toward Th1, Th17, and Th22 cells [172]. Thus, endothelin 1 is possibly involved in the chronic (namely complexed or intermingled) immune response in inflammatory skin diseases [172,174]. In keeping with this notion, topical application of endothelin receptor antagonist was found to attenuate mite-induced dermatitis [62] as well as imiquimod-induced psoriasiform skin inflammation [175]. Notably, there is a mutual feedforward regulatory circuit between IL-25 and endothelin 1. IL-25 up-regulates the production of endothelin 1, while endothelin 1 also up-regulates the production of IL-25 in keratinocytes [119].

## 7. IL-13/IL-4–JAK–STAT6/STAT3 Axis and Oxidative Stress

Oxidative stress is one of the most important cellular reactions in dermatitis [176]. The release of IL-1β from dendritic cells is essential for the efficient induction of hapten-specific T cells [177]. Hapten activates Syk and induces the production of pro-IL-1β [177]. Pro-IL-1β has to be cleaved by caspase 1 to produce mature IL-1β before its release. The hapten-mediated generation of reactive oxygen species (ROS) is indispensable for this caspase 1 activation [177]. In vivo, it has been confirmed that ROS are generated in dermatitis [178]. This in vivo generation of ROS can be visualized by observing the rate of reduction of redox compound tempol signal using dynamic nuclear polarization magnetic resonance imaging (DNP-MRI) [178]. It has been confirmed that mite antigen-induced dermatitis generates a large amount of ROS in local inflamed skin [178].

IL-13/IL-4 activate dual oxidase protein 1 (DUOX1) and generate ROS production [179] (Figure 6). IL-13/IL-4 phosphorylate STAT6, while a protein-tyrosine phosphatase, nonreceptor-type 1 (PTPN1), dephosphorylates p-STAT6 and inhibits STAT6 activation [179,180,181]. ROS inhibit PTPN1 activity and sustain p-STAT6 activity [179,180,181].

When ROS are generated, cells start to operate their antioxidative system to neutralize ROS and avoid damage. Nuclear factor E2-related factor 2 (NRF2) is the master transcription factor for the antioxidative system [31,87,182,183,184,185,186,187,188,189]. Once activated, NRF2 induces various antioxidative enzymes such as NAD(P)H quinone oxidoreductase 1 (NQO1) [31,87], heme oxygenase 1 (HMOX1) [182,183,184,185,186], glutathione peroxidase 2 (GPX2) [187], and superoxide dismutase 2 [188], which neutralize ROS. In general, antioxidative phytochemicals used in folk medicines exert their antioxidative functions via NRF2 activation [189].

IL-13 is also known as a pro-fibrotic cytokine and is believed to be involved in lichenification in AD or tracheal remodeling in asthma [185,190,191] (Figure 5). IL-13 is more pathogenic than IL-4 regarding fibrosis in atopic disorders [192]. IL-13-mediated periostin production is also involved in atopic fibrosis and the periostin production is dependent on IL-13-induced ROS generation [185]. Antioxidative phytochemicals such as cinnamaldehyde actually activate NRF2 and reduce the IL-13-mediated ROS generation and periostin production [185].

## 8. Mechanisms of Action of Pharmaceutical Agents in AD

In the lesional skin of AD, excessive activation of the IL-13/IL-4‒JAK‒STAT6/STAT3 axis inhibits the production of barrier-related proteins, disrupts the barrier function, and induces oxidative stress by ROS generation. Pharmaceutical agents used in standard therapy in AD can inhibit the IL-13/IL-4–JAK–STAT6/STAT3 axis. Corticosteroid [193,194,195], tacrolimus [194,195,196,197], and cyclosporine [193,195,197,198] can inhibit the production of IL-13/IL-4 from immune cells. The reduction of IL-13/IL-4 may attenuate the activity of the IL-13/IL-4–JAK–STAT6/STAT3 axis and decrease ROS generation. It has been shown that 0.05% topical betamethasone dipropionate or 0.05% clobetasol propionate reduces the lesional levels of IL-13, which correlates well with the significant improvement of clinical symptoms in AD [41]. The decrease of IL-13 levels is also significantly associated with the improvement of TEWL, decreased expression of CCL17 and CCL22, and restoration of FLG and LOR expression [41]. Notably, in asthmatic patients, tracheal ILC2s are more tolerant of steroid treatment than circulating ILC2s [199,200].

Dupilumab inhibits the IL-13/IL-4‒JAK‒STAT6/STAT3 axis by interfering with the binding of IL-13/IL-4 and IL-4Rα. Therefore, dupilumab inhibits the elevated production of type 2 chemokines such as CCL17, CCL22, and CCL26 and restores the decreased levels of FLG and LOR [22]. Blockade of the IL-13/IL-4‒JAK‒STAT6/STAT3 axis using either steroid or dupilumab simultaneously decreases the elevated levels of IL-13, IL-31, IL-17A, and IL-22 [22,41]. These results stress that excessive activation of the IL-13/IL-4–JAK–STAT6/STAT3 axis is the fundamental abnormality in AD and that other immunological responses may be secondary to it.

In Japan, the topical pan-JAK inhibitor delgocitinib became commercially available for the treatment of AD in 2020. Delgocitinib can inhibit the activity of JAKs downstream of IL-4Rα/γC, IL-4Rα/IL-13Rα1, and IL-31RA/OSMR and is capable of improving skin eruption, barrier dysfunction, and pruritus [201,202,203,204,205]. All of the standard medicines, steroid, tacrolimus, cyclosporine, dupilumab, and delgocitinib, inhibit the IL-13/IL-4‒JAK‒STAT6/STAT3 axis and subsequent IL-13/IL-4-mediated oxidative stress.

Coal tar and glyteer are historical remedies for treating dermatitis. They activate the AHR axis, up-regulate the expression of FLG, and restore the skin barrier function [31,92]. In general, mere activation of AHR can only restore the FLG expression, but cannot inhibit ROS production. However, medicinal coal tar and glyteer are antioxidative AHR agonists and can activate AHR as well as NRF2 [31,92] (Figure 6). These AHR and NRF2 dual activators can up-regulate various antioxidative enzymes and neutralize the IL-13/IL-4-mediated generation of ROS [92,179,180,181]. A reduction of ROS may also down-regulate STAT3 activation [206,207]. Moreover, the activation of NRF2 can potentially inhibit the function of ILC2 [208].

As coal tar and glyteer are mixtures of various compounds, their therapeutic activity is not consistent. They also have an unfavorable odor. Recently, a single molecular antioxidative AHR agonist, tapinarof 5-[(E)-2-phenylethenyl]-2-[propan-2-yl] benzene-1,3-diol, WBI-1001, GSK2894512, or benvitimod} has attracted particular attention. Tapinarof activates both AHR and NRF2, up-regulates the expression of FLG and IVL, and exhibits antioxidative activity [209,210]. Double-blind clinical trials have shown that topical tapinarof is efficacious for treating patients with AD compared with placebo [211,212,213].

In addition to their effects on keratinocytes, antioxidative AHR ligands can inhibit the phosphorylation of STAT6 and subsequent production of CCL17 and CCL22 in dendritic cells [26]. Antioxidative AHR ligands can also block the enhancing effects of IL-4 on the IL-31-mediated stimulatory function of dendritic cells [214]. Therefore, competitive regulation between the IL-13/IL-4‒JAK‒STAT6/STAT3 axis and the AHR axis is possibly involved in other biological responses. As their mechanistic features are different, combined treatment with AHR/NRF2 dual activators and IL-13/IL-4–JAK–STAT6/STAT3 axis inhibitors may become a more suitable therapeutic approach for AD. In addition, many other biologic and small molecular drugs including anti-IL-13 antibodies and JAK inhibitors are in clinical trials as reviewed elsewhere [215]. Considering the favorable efficacy of some emerging agents, future therapeutic strategy for AD seems quite promising [215].

## 9. Conclusions

In this review article, I mainly describe the regulatory mechanisms of skin barrier-related proteins, focusing on the IL-13/IL-4‒JAK‒STAT6/STAT3 axis and the AHR axis, which are mutually competing systems. Although AD patients are rather heterogeneous, excessive activation of the IL-13/IL-4‒JAK‒STAT6/STAT3 axis and subsequent barrier dysfunction appear to be common and cardinal features in all AD patients. AD appears to be characterized by skin inflammation, barrier dysfunction, and chronic pruritus, which are all attributable to excessive activation of the IL-13/IL-4‒JAK‒STAT6/STAT3 axis and its associated immune reaction. However, it remains unclear whether AHR axis affects pruritus or not. Accumulating evidence underscores the notion that AD can be re-defined as a form of skin inflammation attributable to excessive IL-13/(IL-4) signaling in individuals with atopic diathesis. Targeting the IL-13/IL-4‒JAK‒STAT6/STAT3 axis as well as the AHR axis is a promising strategy to develop new drugs for AD. The combined use of remedies for these two axes may bring further benefits.

## Figures and Tables

**Figure 1 jcm-09-03741-f001:**
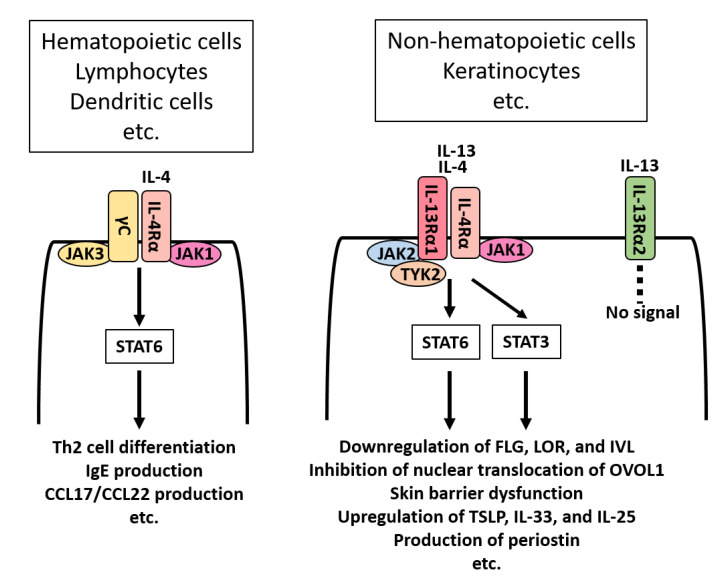
IL-13/IL-4 receptors and their signaling. Hematopoietic cells such as lymphocytes and dendritic cells express IL-4Rα/γC complex. IL-4, but not IL-13, binds IL-4Rα/γC and activates downstream signaling molecules JAK1/JAK3 and thenSTAT6. Activation of the IL-4‒IL-4Rα/γC‒JAK1/JAK3‒STAT6 axis induces Th2-deviated T-cell differentiation, IgE production in B cells, and the production of Th2 chemokines such as CCL17 and CCL22 from dendritic cells. On the other hand, non-hematopoietic cells such as keratinocytes express IL-4Rα/IL-13Rα1 complex. Both IL-13 and IL-4 bind IL-4Rα/IL-13Rα1 and activate downstream JAK1/TYK2/JAK2 and then STAT6/STAT3. Activation of the IL-13/IL-4‒IL-4Rα/IL-13Rα1‒JAK1/TYK2/JAK2‒STAT6/STAT3 axis down-regulates FLG, LOR and involucrin (IVL) expression, inhibits the nuclear translocation of OVO-like 1 (OVOL1), disrupts the skin barrier function, and up-regulates the production of thymic stromal lymphopoietin (TSLP), IL-25, and IL-33 in keratinocytes. Keratinocytes also express IL-13Rα2, which is a decoy receptor for IL-13. IL-13Rα2 binds IL-13 with high affinity, but does not signal.

**Figure 2 jcm-09-03741-f002:**
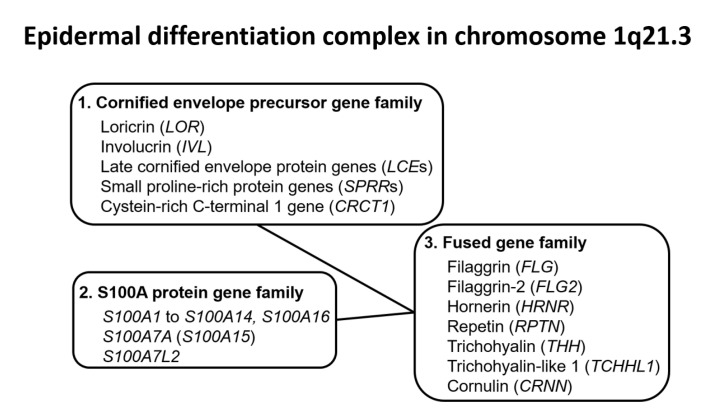
Genes in the epidermal differentiation complex located on chromosome 1q21.3. Most genes encoding barrier-related proteins are located in the epidermal differentiation complex on chromosome 1q21.3. The epidermal differentiation complex includes cornified envelope precursor gene family members such as *LOR* and *IVL*, S100A protein gene family members such as *S100A7*, and fused gene family members such as *FLG* and *FLG2*. The fused gene family was generated by fusion of the cornified envelope precursor gene and the S100A protein gene.

**Figure 3 jcm-09-03741-f003:**
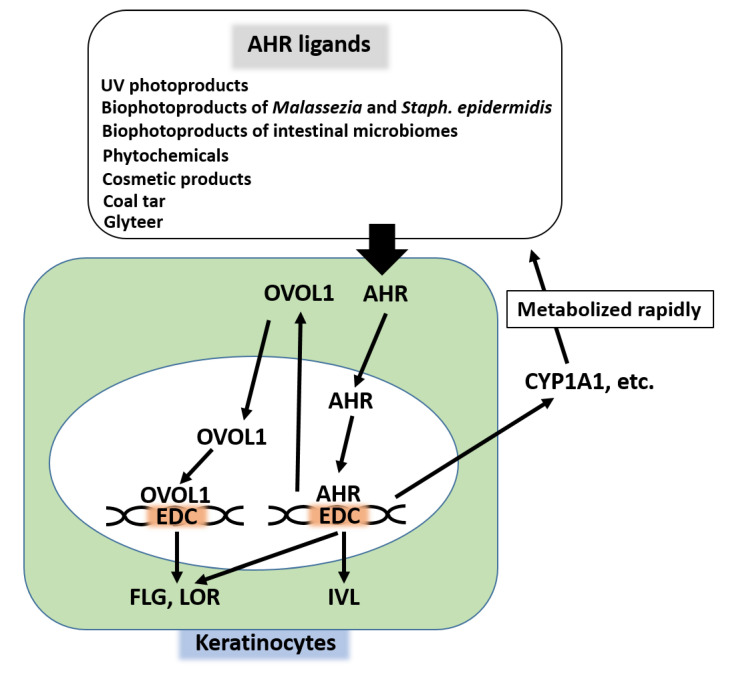
Up-regulation of FLG, LOR, and IVL by the AHR axis. AHR is activated by various ligands such as ultraviolet ray (UV) photoproducts, bioproducts of *Malassezia* and *Staphylococcal epidermidis*, bioproducts of intestinal microbiomes, phytochemicals, cosmetic products, and medicinal coal tar and glyteer. Once activated, cytoplasmic AHR translocates into the nucleus, binds EDC, and up-regulates the expression of FLG, LOR, and IVL. AHR also up-regulates the expression of OVOL1. Cytoplasmic OVOL1 translocates into the nucleus, binds EDC, and up-regulates FLG and LOR. AHR also up-regulates the expression of xenobiotic-metabolizing enzymes such as cytochrome p450 1A1 (CYP1A1), which rapidly metabolize the AHR ligands. Thus, homeostatic regulation of barrier-related proteins is operated by the AHR axis.

**Figure 4 jcm-09-03741-f004:**
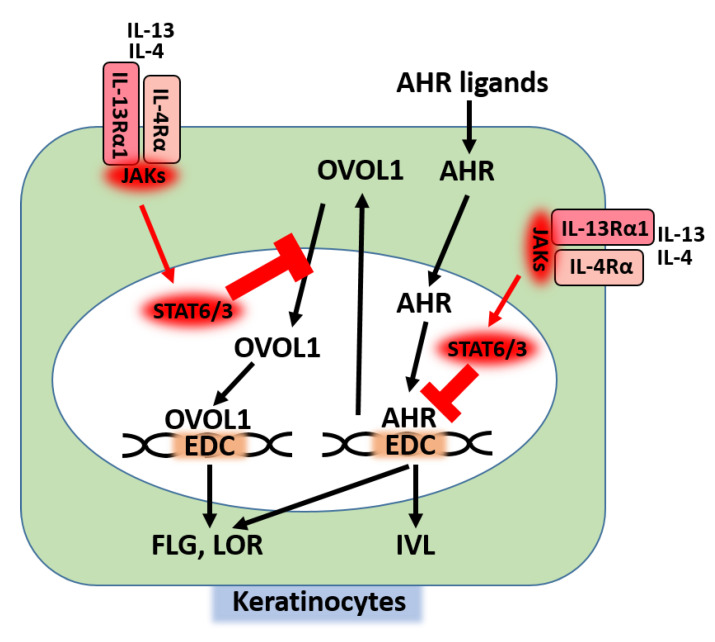
Inhibitory action of the IL-13/IL-4‒JAK‒STAT6/STAT3 axis on the AHR axis. The IL-13/IL-4‒JAK‒STAT6/STAT3 axis inhibits the AHR-mediated transcription of *FLG*, *LOR*, and *IVL*. The IL-13/IL-4‒JAK‒STAT6/STAT3 axis also inhibits the cytoplasmic-to-nuclear translocation of OVOL1.

**Figure 5 jcm-09-03741-f005:**
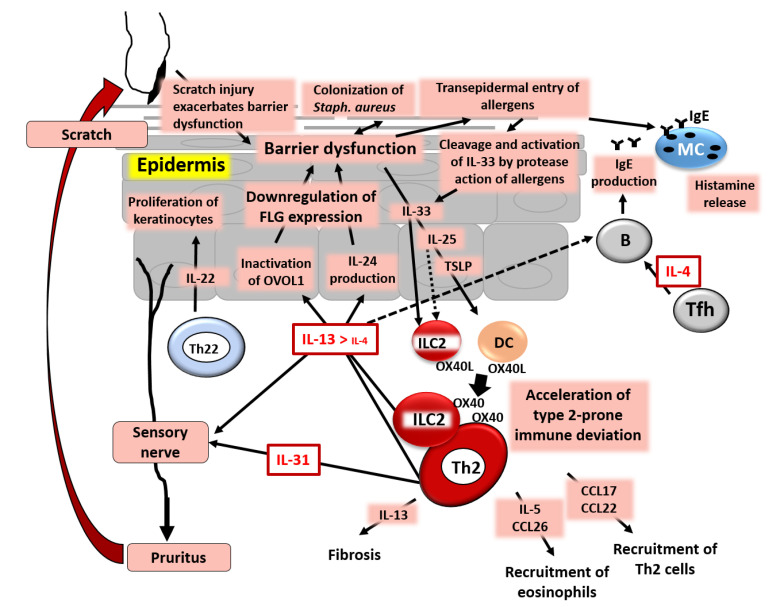
Simplified pathogenesis of AD. IL-4 and especially IL-13, produced from group 2 innate lymphoid cells (ILC2s) and T helper 2 (Th2) cells, down-regulate FLG expression and induce barrier dysfunction via inactivation of OVOL1. IL-13/IL-4 stimulate keratinocytes to produce periostin and then IL-24, which also down-regulates FLG expression. Keratinocytes in the barrier-disrupted epidermis release thymic stromal lymphopoietin (TSLP), IL-25, and IL-33, which up-regulate the expression of OX40L in ILC2s and dendritic cells (DCs). OX40L-positive ILC2s and DCs accelerate the differentiation of OX40-positive ILC2s and Th2 cells. The barrier dysfunction triggers the colonization of *Staphylococcus aureus* and transepidermal entry of allergens. Many allergens have protease activity, which cleaves full-length IL-33 to active short-form IL-33. T follicular helper (Tfh) cells produce IL-4, which stimulates B cells to produce IgE. IgE on mast cells (MCs) is ligated by allergens and the MCs then release histamine and other chemical mediators. IL-31 and IL-13/IL-4 stimulate sensory nerves and induce pruritus with subsequent scratch behavior, which further exacerbates skin barrier dysfunction. An IL-13/IL-4-rich milieu up-regulates the production of CCL17 and CCL22, which induce the preferential recruitment of Th2 cells. An IL-13/IL-4-dominant microenvironment also up-regulates the production of IL-5 and CCL26, which attract eosinophils. IL-13 is responsible for the pro-fibrotic process and is also involved in lichenification and tissue remodeling.

**Figure 6 jcm-09-03741-f006:**
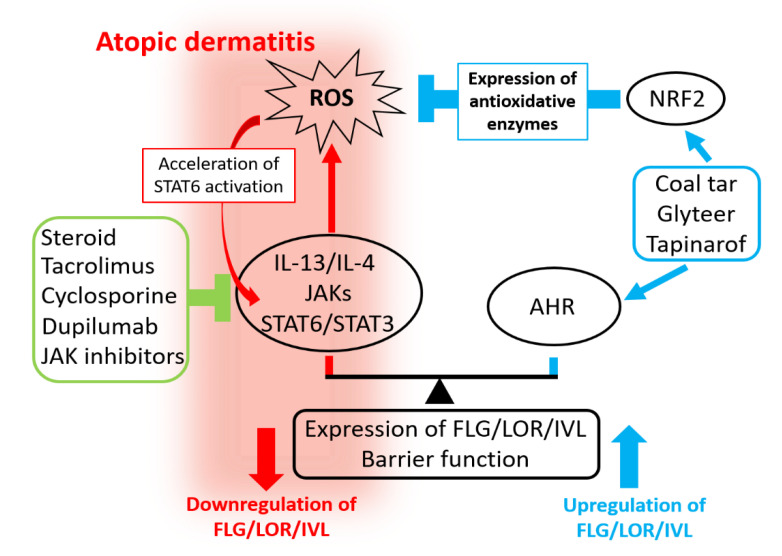
Mechanisms of action of therapeutic remedies for AD. Skin barrier function is regulated via competition between the AHR axis (up-regulation of barrier) and the IL-13/IL-4‒JAK‒STAT6/STAT3 axis (down-regulation of barrier). The IL-13/IL-4‒JAK‒STAT6/STAT3 axis down-regulates the expression of FLG, LOR, and IVL and disrupts barrier function. It also generates reactive oxygen species (ROS), which accelerate the STAT6 activation. Standard and current therapeutic agents inhibit the IL-13/IL-4‒JAK‒STAT6/STAT3 axis. Steroid, the calcineurin inhibitor tacrolimus, and cyclosporine inhibit the production of IL-13/IL-4 from Th2 cells and ILC2s. The anti-IL-4Rα antibody dupilumab inhibits the IL-13/IL-4 ligation. JAK inhibitors inhibit the activation of JAK. The AHR axis up-regulates the expression of FLG, LOR, and IVL and strengthens barrier function. Antioxidative AHR agonists activate both AHR and nuclear factor E2-related factor 2 (NRF2). Coal tar, soybean tar glyteer, and tapinarof are antioxidative AHR agonists. They can up-regulate the expression of FLG, LOR, and IVL and strengthen the barrier function via AHR. They can also neutralize ROS by antioxidative enzymes induced by NRF2 activation. Combined treatments targeting the IL-13/IL-4–JAK–STAT6/STAT3 axis and the AHR axis may enhance the treatment efficacy.

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
