# Peer review of "Regulation of Skin Barrier Function via Competition between AHR Axis versus IL-13/IL-4‒JAK‒STAT6/STAT3 Axis: Pathogenic and Therapeutic Implications in Atopic Dermatitis"

_jcm, 2020, doi:10.3390/jcm9113741_

Round 1

Reviewer 1 Report

Dear author

Thank you for a very comprehensive and interesting dissection og the mechanisms of AD.

The manuscript is a comprehensive, well-written review on the pathomechanism and itch mechanistic factors of Atopic Dermatitis. Though nothing new is brought to the field of AD research, the paper does however use a very concise and detailed presentation of the pathways involved in barrier function and skin inflammation.

The pathogenic differences of AHR and JAK/STAT axis on itch in AD and how it implicates therapeutic alternatives.

The paper is highly relevant, though not original, as it is a review, referencing numerous articles published by the author.

The paper adds an overview of published literature, although not exhaustively.

It is well written and in academic english.

The text is very detailed but the formulations are clear.

The conclusion summarizes the main points of the manuscript and throws perspective to future therapeutic options in AD.

They do address the main question posed.

Author Response

Reply to the Reviewer 1

The manuscript is a comprehensive, well-written review on the pathomechanism and itch mechanistic factors of Atopic Dermatitis. Though nothing new is brought to the field of AD research, the paper does however use a very concise and detailed presentation of the pathways involved in barrier function and skin inflammation.

→ Thank you very much for your encouraging comments.

The pathogenic differences of AHR and JAK/STAT axis on itch in AD and how it implicates therapeutic alternatives.

→ Thank you very much for your important comment. The relationship between AHR and pruritus is still unclear. According to your comment, we added the following sentence in Conclusion. “However, it remains unclear whether AHR axis affects pruritus or not.” (Line 523-524)

The paper is highly relevant, though not original, as it is a review, referencing numerous articles published by the author.

→ Thank you very much for your comment. As the Publishing Editors request more than 5000 words for review article, the article is long. Please understand the situation.

The paper adds an overview of published literature, although not exhaustively.

→ Thank you for your comment.

It is well written and in academic english.

→ Thank you for your comment.

The text is very detailed but the formulations are clear.

→ Thank you very much for your comment.

The conclusion summarizes the main points of the manuscript and throws perspective to future therapeutic options in AD.

→ Thank you for your favorable comment.

They do address the main question posed.

→ Thank you for your comment.

Thank you very much again for your very helpful comments. I hope the revised article is now suitable for publication in JCM.

Reviewer 2 Report

In the current study, the author provided significant insight into the potential signaling involved and possible treatment of atopic dermatitis (AD). IL-13/IL-4‒JAK‒STAT6/STAT3 as well as the AHR axis are the novel and principal pathways that promote skin inflammation, skin barrier dysfunction, chronic pruritus and oxidative stress, targeting which could be a promising strategy to develop new drugs for AD. This review is clear and well written.

Minor Comments:

The statements in lines 51 to 59 are repeated in lines 66 to 76. These statements should be modified or shortened to avoid repetition.

Line 346, “It is possible that the blockade of itch-scratch behavior by dupilumab attenuates the infiltrations of Th17/Th22 cells”, should be modified as: ” It is possible that the blockade of itch-scratch behavior by dupilumab may attenuate…..”

Major Comments:

The author should include the epidemiology of AD in different age groups in the introduction section

Research showed that dupilumab not only reduces Th2 associated molecules such as CCL17, CCL18 and CCL26, but also strongly decreased mediators associated with Th17 and Th22 responses, such as S100A proteins, PI3/elafin and IL-23p19 showing Th17/IL-23 axis is upregulated in AD patients and have a role in AD development. In this current review, the author should discuss Th17/IL-23 axis in AD.

The author should discuss about, treatment of AD by IL-22 blockade with fezakinumab in line 355. Include citation from, “Brunner PM et al, 2019, Baseline IL-22 expression in patients with atopic dermatitis stratifies tissue responses to fezakinumab”

The author should discuss in detail regarding AD therapeutic strategies using humanized monoclonal antibodies targeting different cytokines (e.g. Lebrikizumab, Tralokinumab, Fezakinumab, Ustekinumab) or JAK1 and  NK-1 inhibitors like RPF-04965842, Tradipitant.

Staphylococcal superantigens are majorly involved in the pathogenesis of itch in AD where IL-31 negatively affects the production of AMPs and regulates intense pruritus. Moreover, S. aureus superantigens resistant to glucocorticoid and tacrolimus made AD therapy more challenging. The author should include these points in the current review.

Author Response

Reply to the Reviewer 2

In the current study, the author provided significant insight into the potential signaling involved and possible treatment of atopic dermatitis (AD). IL-13/IL-4‒JAK‒STAT6/STAT3 as well as the AHR axis are the novel and principal pathways that promote skin inflammation, skin barrier dysfunction, chronic pruritus and oxidative stress, targeting which could be a promising strategy to develop new drugs for AD. This review is clear and well written.

→ Thank you very much for your encouraging comments.

Minor Comments:

The statements in lines 51 to 59 are repeated in lines 66 to 76. These statements should be modified or shortened to avoid repetition.

→ Thank you very much for your comment. This is an editorial mistake. Line 66 to 76 is a Figure legend, so it must be smaller sized character. I collected the size and I also amended the repetition of abbreviations.

Line 346, “It is possible that the blockade of itch-scratch behavior by dupilumab attenuates the infiltrations of Th17/Th22 cells”, should be modified as: ” It is possible that the blockade of itch-scratch behavior by dupilumab may attenuate…..”

 → Thank you very much for your comment. According to your comment, I amended the sentence.

Major Comments:

The author should include the epidemiology of AD in different age groups in the introduction section

→ Thank you for your helpful comment. According to your comment, we added the following sentences in the Introduction.

Line 32 to 35  “The patients with AD show heterogeneous clinical presentation and at least 6 subtypes are defined by their natural course with the early-onset-early-resolving type being the most frequent one [4,5]. However, subsequent recurrence is also frequently seen and the recurrent symptoms are usually more indolent than the initial ones [4,5].”

Research showed that dupilumab not only reduces Th2 associated molecules such as CCL17, CCL18 and CCL26, but also strongly decreased mediators associated with Th17 and Th22 responses, such as S100A proteins, PI3/elafin and IL-23p19 showing Th17/IL-23 axis is upregulated in AD patients and have a role in AD development. In this current review, the author should discuss Th17/IL-23 axis in AD.

→ Thank you very much for your critical comment. According to your comment, we added the following sentences.

Line 365 to 372    “In addition, some patients actually manifest psoriasiform eruption with IL-23/IL-17A overexpression during dupilumab therapy, suggesting that IL-23/IL-17A axis is potentially active and meaningful in these particular AD patients [149]. Multipolarity of these cytokines in AD may explain the heterogeneity and age/race differences of AD [150]. On the other hand, blockade of the IL-13/IL-4 signal by dupilumab alone was shown to normalize the increased IL-17A/IL-22 signals including S100A proteins, elafin and IL-23p19 [22], which may support the notion that excessive IL-13/IL-4 signal also causes the elevation of IL-17A/IL-22.”

The author should discuss about, treatment of AD by IL-22 blockade with fezakinumab in line 355. Include citation from, “Brunner PM et al, 2019, Baseline IL-22 expression in patients with atopic dermatitis stratifies tissue responses to fezakinumab”

→ Thank you very much for your helpful comment. According to your comment, we added the following sentences adding the Brunner’s article.

Line 361 to 363  “The anti-IL-22 antibody fezakinumab is more efficacious in severe than in mild AD [145] and in patients with high rather than low serum IL-22 levels [146].”

The author should discuss in detail regarding AD therapeutic strategies using humanized monoclonal antibodies targeting different cytokines (e.g. Lebrikizumab, Tralokinumab, Fezakinumab, Ustekinumab) or JAK1 and NK-1 inhibitors like RPF-04965842, Tradipitant.

→ Thank you very much for your helpful comment. According to your comment, we added the following sentences.

Line 511 to 514  “In addition, many other biologic and small molecular drugs including anti-IL-13 antibodies and JAK inhibitors are in clinical trials as reviewed elsewhere [217]. Considering the favorable efficacy of some emerging agents, future therapeutic strategy for AD seems quite promising [217]”

Staphylococcal superantigens are majorly involved in the pathogenesis of itch in AD where IL-31 negatively affects the production of AMPs and regulates intense pruritus. Moreover, S. aureus superantigens resistant to glucocorticoid and tacrolimus made AD therapy more challenging. The author should include these points in the current review.

→ Thank you very much for your helpful comments. According to your comment, we added the following sentences in the revised paper.

Line 156 to 160  “Colonization of Staphylococcus aureus upregulates the expression of antimicrobial peptides such as human β-defensin 2 and -3 and RNAse7, however, IL-31 inhibits their expression and may accelerate the Staphylococcal infection [73]. Staphylococcal superantigen may also induce glucocorticoid insensitivity by inhibiting the nuclear translocation of glucocorticoid receptor α [74].”

Thank you very much again for your very helpful comments. I hope the revised article is now suitable for publication in JCM.

Reviewer 3 Report

This is a very well-written, extremely thorough review that I very much enjoyed. My only thought is that it is so thorough and so long, that it feels like a textbook chapter at times. I think it would benefit greatly from significant shortening if possible, as it is very difficult to read in one sitting right now.

It may even be better to break it up into 2 or 3 smaller reviews.

The biggest issue here is that it's not really focusing on what was stated in the title, it goes into many other areas.

Specifics:

Line 30: "reduce treatment adherence"? Satisfaction and quality of life, yes, but why do these reduce adherence to treatments?

Line 31: This paper is very long, I'm not sure we need to get into defining "atopy" and certainly don't think we need to discuss IgE here. 

Line 101: Do we need to get into the failed mepolizumab trial? I just feel like I'm drowning in information. A more targeted approach would be higher-yield.

Line 121: You (rightly) state that anti-histamines do not help with the itch of AD, but then use an unusual reference: 55. Kawashima et al. Addition of fexofenadine to a topical corticosteroid reduces the pruritus associated with atopic dermatitis in a 1-week randomized, multicentre, double-blind, placebo-controlled, parallel-group study. Br. J. Dermatol. 2003, 148, 1212-1221. 

This reference concludes: "Fexofenadine HCl 60mg twice daily demonstrated a rapid, significant improvement in the pruritus associated with atopic dermatitis, with a safety profile equivalent to that of placebo." 

Maybe picking a different reference here? Or just deleting this section? 

Line 148: We're using LOR and FLG as abbreviations for the proteins, but these are the genes as well, so this is confusing. 

Line 182: You've already told us the abbreviation for AHR in the last paragraph--I think this is happening a lot in the document which makes it seem even busier and more complex.

Paragraph starting at 285: Not sure why we need this paragraph.

Author Response

Reply to Reviewer 3

This is a very well-written, extremely thorough review that I very much enjoyed. My only thought is that it is so thorough and so long, that it feels like a textbook chapter at times. I think it would benefit greatly from significant shortening if possible, as it is very difficult to read in one sitting right now. It may even be better to break it up into 2 or 3 smaller reviews. The biggest issue here is that it's not really focusing on what was stated in the title, it goes into many other areas.

→ Thank you very much for your encouraging and helpful comments. As the Publishing Editors request more than 5000 words for review article, the article is long. Please understand the situation.

Specifics:

Line 30: "reduce treatment adherence"? Satisfaction and quality of life, yes, but why do these reduce adherence to treatments?

→ Thank you very much for your helpful comment. I agree with your comment. According to your comment, we amended the sentences as follows.

Line 36 to 38;  “which reduce quality of life and treatment satisfaction among afflicted patients [6-8]. Their adherence to treatment is also markedly deteriorated [9-11].”

Line 31: This paper is very long, I'm not sure we need to get into defining "atopy" and certainly don't think we need to discuss IgE here. 

→ Thank you very much for your helpful comment. I agree with your comment. According to your comment, we deleted these sentences in the revised article.

Line 101: Do we need to get into the failed mepolizumab trial? I just feel like I'm drowning in information. A more targeted approach would be higher-yield.

→ Thank you very much for your helpful comment. I agree with your comment. According to your comment, we deleted these sentences in the revised article.

Line 121: You (rightly) state that anti-histamines do not help with the itch of AD, but then use an unusual reference: 55. Kawashima et al. Addition of fexofenadine to a topical corticosteroid reduces the pruritus associated with atopic dermatitis in a 1-week randomized, multicentre, double-blind, placebo-controlled, parallel-group study. Br. J. Dermatol. 2003, 148, 1212-1221.  This reference concludes: "Fexofenadine HCl 60mg twice daily demonstrated a rapid, significant improvement in the pruritus associated with atopic dermatitis, with a safety profile equivalent to that of placebo."  Maybe picking a different reference here? Or just deleting this section? 

→ Thank you very much for your helpful comment. I agree with your comment. According to your comment, the cited paper was substituted for the following article.

Matterne, U.; Böhmer, M.M.; Weisshaar, E.; Jupiter, A.; Carter, B.; Apfelbacher, C.J. Oral H1 antihistamines as 'add-on' therapy to topical treatment for eczema. Cochrane Database Syst. Rev. 2019, 1, CD012167.

Line 148: We're using LOR and FLG as abbreviations for the proteins, but these are the genes as well, so this is confusing. 

→ Thank you very much for your helpful comment. I agree with your comment. According to your comment, I checked them carefully and amended them. Italic FLG, LOR and IVL mean genes and non-italic mean proteins.

Line 182: You've already told us the abbreviation for AHR in the last paragraph--I think this is happening a lot in the document which makes it seem even busier and more complex.

→ Thank you very much for your helpful comment. I checked and amended them carefully.

Paragraph starting at 285: Not sure why we need this paragraph.

→ Thank you very much for your helpful comment. According to your comment, we deleted the unnecessary sentences.

Thank you very much again for your very helpful comments. I hope the revised article is now suitable for publication in JCM.

Round 2

Reviewer 3 Report

This is a very well-written and extremely thorough. I do think the changes make it a stronger paper.